# Type and amount of help as predictors for impression of helpers

**Arvid Erlandsson** [ID]*, **Mattias Wingren, Per A. Andersson**

Department of Behavioral Sciences and Learning, Linköping University, Linköping, Sweden

* arvid.erlandsson@liu.se

## Abstract

Impression of helpers can vary as a function of the magnitude of helping (amount of help) and of situational and motivational aspects (type of help). Over three studies conducted in Sweden and the US, we manipulated both the amount and the type of help in ten diverse vignettes and measured participants' impressions of the described helpers. Impressions were almost unaffected when increasing the amount of help by 500%, but clearly affected by several type of help-manipulations. Particularly, helpers were less positively evaluated if they had mixed motives for helping, did not experience intense emotions or empathy, or if helping involved no personal sacrifice. In line with the person-centered theory of moral judgment, people seem to form impressions of helpers primarily based on the presumed underlying processes and motives of prosociality rather than its consequences.

**Data Availability Statement:** The data and code for all studies have been uploaded at the Open Science Framework https://osf.io/v2jp9/?view_only=9a5b52fc7d29461096644ce136a16aa9.

## Introduction

Moral (and immoral) behavior rarely involves only the actor and the recipient but often also one or more observers [1]. Observers quickly form impressions about warmth and competence, but more importantly about actors' moral character [2]. Those who harm others are perceived as bad whereas those who help others are perceived as good, but situational aspects and perceived underlying motivations can influence both negative and positive impressions. Past research has primarily focused on how people perceive others who engage in harmful behavior or make decisions in sacrificial dilemmas [3–11]. Some, but considerably less, research have focused on how people perceive those who behave extraordinary moral (see e.g. [12]). In this paper we are interested in which aspects that best predict impressions of helpers.

### Person-centered morality

The theoretical starting point of this paper is the person-centered approach to moral judgment [13–17], which argues that people, when making moral judgments, are intuitive virtue ethicists and care more about the character of individuals (e.g. is Jimmy a good or bad person?), than about the morality of the acts (e.g. is this behavior morally right or wrong?). The person-centered approach was created as a response to earlier models of morality, which investigated human morality primarily by having people evaluate the acceptability of specific acts such as lying and stealing [18], or killing one to save many [19]. Importantly, the person-centered approach does not deny that actions and decisions are important for moral judgments. An

**Funding:** This work was supported by the Swedish Research Council under Grant number 2017-01827 (AE). https://www.vr.se/english The funders had no role in study design, data collection and analysis, decision to publish, or preparation of the manuscript.

**Competing interests:** The authors have declared that no competing interests exist.

atypical act from a person you know nothing about can be very informative of that person's moral character. Briefly explained, people use observed moral behavior together with past experiences and situational cues to infer the motives underlying the behavior, which in turn reflects the actor's moral character and predict how that person will behave in the future.

What kind of behaviors influence character evaluations the most? Two robust finding is that extreme behavior (e.g. killing) is more diagnostic than moderate behavior (e.g. lying), and that immoral behavior (e.g. taking money) is more diagnostic than moral behaviors (e.g. giving money; [20]). It takes many positive behaviors to make others perceive you as a good person, but it takes few negative behaviors to make them see you as a bad person. Much research has investigated this positive-negative asymmetry [21–23], and much other research have focused specifically on how different negative behaviors influence person perception [24–26]. In this paper, we focus on moral praise, which has received considerably less attention than moral blame [27], and investigate which aspects of helping behaviors that elicit more or less positive evaluations.

We test this experimentally by using multiple vignettes where we describe hypothetical helpers (individuals or corporations that engage in prosocial behavior), and ask participants to evaluate the helpers' character. In each vignette, we factorially vary the "amount of help", and one out of ten different "types of help".

## Amount of help as a predictor for positive evaluations?

According to the normative theory of Utilitarianism, consequences (direct or indirect) are the only thing that should matter when evaluating moral behavior. The influential "effective altruism" movement apply utilitarian theory specifically on helping behavior and suggests that much of our helping today is ineffective and/or symbolic and that we are morally indebted to change this [28,29]. Effective altruists argue that in order to do the most good, people should avoid spending time on volunteering or local helping initiatives, but rather aim for a well-paid job and then donate a large proportion of their salary to effective charitable organizations that focus on causes where it is possible to do the most good per dollar.

Likewise, effective altruists are less likely to see helping as something categorical (non-helping vs. helping) but rather as something continuous where the value of helping increases linearly with the amount of good that is made [30]. Everything else equal, an effective altruist should, in theory, be ten times more motivated to help when it is possible to save 10 persons than when it is possible to save one person. Using the same logic for impression formation, an effective altruist would, in theory, praise a helper who donates $50, volunteers 20 hours or saves 10 lives, five times more than a helper with an identical background who in the same situation donates $10 (to the same cause), volunteers 4 hours or saves 2 lives [31]. In short, if people were effective altruists, amount of help would influence evaluation of helpers.

Despite this, much research suggests that people are insensitive to amounts in helping situations. Scope-insensitivity refers to the human inability to adjust one's compassionate behavior as the amount of need (e.g. number of people at risk) or the amount of good one can do (e.g. number of people possible to help) increases [32–36]. This implies that people are equally (and sometimes more) motivated to help when they can help fewer people as when they can help more people. For example, one predetermined recipient elicits more help than several undetermined recipients [37], and people are more motivated to help when they can save 100% of 10 people in need than when they can save 80% of 20 people in need—despite that this implies that six fewer lives are saved [38–42].

Although scope-insensitivity has been frequently investigated by using helping behavior or self-rated helping motivation as the outcome variable, there has not been as much research on scope-insensitivity in the impression formation context. One exception is Krull, Seger and

Silvera [43] who manipulated the amount of help and the "willingness to help" and found that amount of help only predicted impressions if the help was given willingly (e.g. while smiling). Also, research investigating cooperation in economic games suggest that evaluations are no more positive when people help more rather than less [44–46], and that we sometimes want to avoid overly generous people as we believe they establish undesired norms [47]. In a recent opinion article [27], Anderson et al. suggest that moral praise is less magnitude-dependent than moral blame, meaning that harming much is considered worse than harming some whereas helping much is considered equally good as helping some. In this article, we empirically test how an increase in the amount of help (e.g. ×5) influences how helpers are evaluated.

## Ten types of helping

Whereas scope-insensitivity concerns the lack of effect when the amount of need or help increase, there are several other aspects that have been suggested and sometimes shown to influence responses in helping situations. Rather than focusing on a single aspect of a helping situation, this paper approaches this research question broadly and investigate ten different (but to some extent overlapping) aspects–each in a separate vignette. In the first three vignettes, we manipulate the motivation underlying the helping behavior directly whereas the last seven vignettes manipulate situational factors of the helping which can be used to make inferences about the helper's underlying motives.

These ten aspects will be collectively referred to as *types of help*, but discussed separately, and each type of help-manipulation is orthogonal to the amount of help-manipulation. The ten included type of help-manipulations were selected based on existing prosociality research and by personal experiences, and chosen to represent a wide array of motivational and situational aspects that can influence impression formation of helpers. They are not intended to illustrate a comprehensive list. Variations of some of the included manipulations have been included in previous research, but other manipulations have, as far as we know, not been tested in an impression formation context previously (see below).

**Experiencing vs. not experiencing emotions (motivational).** Some research suggests that people who experience (and express) intense emotions in social situations are seen more favorably than those who do not. People like helpers more if they express positive affect than if they are neutral or express negative affect [43,48], and helpers motivated by a perceived obligation or by cost-benefit thinking are perceived less positively than helpers who are emotionally motivated (especially so for low-cost helping; [49]). Also, helpers who experience emotions before (sympathy) and after (warm glow) making helping decisions are perceived as more honest and moral than those who do not [50]. In our "emotional reactions vignette" we test if emotionally touched helpers who give a low amount are perceived more positively than helpers who are not emotionally touched but give a high amount.

**Motivated by empathy vs. distress (motivational).** Experiencing emotions is important but in order to be perceived as fully moral we might have to be motivated by the right type of emotion. In a long-lasting debate about altruistically and egoistically motivated helping, many social psychologists made a distinction between personal distress (helping in order to relieve one's negative emotional state, e.g. guilt) and empathy (helping in order to relieve the suffering for someone else; [51,52]. Distress is an egoist emotion, whereas empathy is an altruistic emotion [53]. In our "empathy vignette" we test if empathy-motivated low-amount helping is perceived more positively than distress-motivated high-amount helping. We do not know of any study that investigated this previously.

**Pure vs. mixed motives (motivational).** Helping that is perceived to be motivated partly by anticipated personal benefits (e.g. volunteering in order to be close to an attractive person)

is less approved of than helping that is motivated by purely other-focused motives, and this seems to be driven by different counterfactuals (e.g. he would not have volunteered if she wasn't there; [54,55]). Observers sometimes perceive a win-win situation (e.g. a strategy that is good both for business and for the environment) as no more moral than profit-seeking strategies with no environmental benefits [56], and helping that is motivated by material or social benefits is perceived as no more altruistic than non-helping [57]. In our "non-tainted altruism vignette" we test if short-time volunteering motivated only by "doing something for one's community" (a pure motive), is perceived more positively than long-time volunteering motivated by an additional desire to spend time with a romantic crush (mixed motives).

**Helping vs. not helping identified victims (situational).**   The identifiable victim effect is one of the most famous helping effects (albeit not one of the most robust; [58–60]), and predicts that people will help more when they can help a single identified innocent person in need. In an impression formation-context, one study found that immoral behavior was more harshly evaluated when the harmful consequences for an identified beneficiary was emphasized [61]. In our "identified victims vignette" we test if low-amount donors who aid individual beneficiaries are perceived more positively than high-amount donors who aid only statistical beneficiaries. To our knowledge, this has not been tested previously.

**Direct vs. indirect helping (situational).**   Volunteering is a form of direct helping whereas donating to a charitable organization is a form of indirect helping. Research suggest that direct actions are more diagnostic of a person's moral character, e.g. indirect harm is less criticized than direct harm [62]. One study found that philanthropists who helped directly (e.g. by doing dental work on needy people) were perceived as less selfish than those who helped indirectly by donating money [63]. Further, Johnson and Park [64] found that direct helpers (giving time) were perceived as more praiseworthy than indirect helpers (giving money), whereas Reed II, Aquino and Levy [65] found that people who highly value a moral identity prefer to help directly with time whereas those who do not, prefer indirect donations. In our "directness vignette" we test if direct helping (volunteering) that saves fewer lives is perceived more positively than indirect helping (donating money) that saves more lives.

**Costly vs. costless helping (situational).**   Helping that is psychologically more costly (e.g. more demanding, distressing or painful) is usually perceived as more moral than less costly helping, even when the consequences for the beneficiaries are exactly the same [66]. For example, Olivola and Shafir [67] found that people were more willing to sponsor a friend's fundraising effort if the fundraising involved some pain (e.g. the ice-bucket challenge), and Leliveld and Bolderdijk (working paper) found that people often have a negative impression of those who gain financially while raising money for charity. In our "personal sacrifice vignette" we test if a volunteer doctor who saves few lives while living in the slums is perceived more positively than a helper who saves many lives while living in an affluent area.

**Public vs. private helping (situational).**   Helping can sometimes be kept private or made public. Although making one's prosocial act public can inform others of your moral character, the very act of actively publicizing it can signal that you are selfishly motivated [68–70]. This implies that publicizing ones helping (i.e. bragging) can undermine the information you are trying to convey (that you are a good person). Research by Monin [71,72] suggests that people sometimes are threatened by other's publicly displayed moral behavior, and regulate this by questioning the helper's motives. In our "keeping help private vignette" we test if low-amount helping that is kept private is perceived more positively than high-amount helping that is intentionally made public.

**Matching vs. surpassing others helping (situational).**   Social norms can strongly motivate helping, meaning that people help more when they think that others help [73], and when they believe that helping is approved of. Whereas donating in line with the social norm (i.e.

matching others helping) is done to avoid being seen as immoral, it is also possible to signal that one is morally or economically superior by doing more than ones fair share (i.e. surpassing others' helping). People in general (but especially men in the presence of attractive females) sometimes engage in this type of competitive prosociality [74,75]. In the "matching other's donation vignette" we test if a person who donates a low amount and matches a friend's donation is perceived more positively than a person who donates a high amount and surpasses the friend's donation. We think we are the first to test this in an impression formation context.

**Equal vs unequal allocations (situational).** Sometimes helpers must distribute resources across beneficiaries. People generally value fairness, equal treatment and justice, and although there are many ways to understand these constructs, unequal helping allocations (e.g. spending all of one's available resources on a single cause) is more morally ambiguous and less preferred than an equal allocation (e.g. splitting the resources evenly across all causes; [76]. Participants in [77] made choices between an equitable option (minimizing inequalities) or an effective option (maximizing the amount of good) and it was found that peoples' preferences differed as a function of which option that was labeled the moral choice, suggesting that unequal allocators might seem less sympathetic mostly because they refuse some helping-requests. In our "equal helping vignette" we test if a lottery winner who donates a low amount to charity but splits it across all requesting organizations is perceived more positively than one who donates a high amount, but give everything to a single organization. We believe that this has not yet been investigated in an impression formation context.

**Upward vs. downward donation adjustments (situational).** Human perception is much more sensitive to directional changes than to absolute levels and this can lead to a small but increasing societal problem being seen as worse and more worthy of attention than a larger but decreasing societal problem [78], because people tend to use the previous estimate as an anchor [79]. We here test this effect in an impression formation context for the first time. Just as sports teams are allegedly evaluated primarily based on their latest game, helpers might be evaluated primarily on their latest moral decision. In our "changing amount vignette" we test if a helper who increases the monthly donation to $15 is perceived more positively than a helper who decreases the monthly donation to $50.

## The current studies

We conducted three empirical studies that investigated to what extent the amount and type of helping influence impressions of helpers when tested between groups. Studies 1a and 1b used identical designs and similar vignettes (six vignettes in Study 1a, all ten in Study 1b), but were conducted in different countries (Sweden and USA), and on different platforms (paper & pen and Amazon Mechanical Turk). Study 2 tested four of the ten original vignettes both when the helper was an individual and when it was a corporation (to explore whether the experimental effects of type and amount of help on impressions are similar or different for individuals and groups [80]. Further, Study 2 tested a potential way to make people base more of their impressions on the amount of helping, by adding a utility-reminder question to make the amount of help more salient for participants.

It should be pointed out that our studies are largely exploratory and conducted with a wide rather than narrow focus, and this makes this paper's contribution general rather than specific. We have purposely chosen to include multiple vignettes (each manipulating a situational or motivational aspect related to helping), rather than concentrating all effort on a specific aspect. This paper contributes to the collective knowledge, as it reports rigorous basic experimental social psychology research that is theoretically grounded in the person-centered approach to moral judgment [13,14], and can be applied to societal issues (e.g. effective altruism). It also

contributes by merging two research fields (moral impression formation and scope-insensitivity in helping situations), and will hopefully provide an empirical starting point for future research on this amalgamation.

## Study 1

Study 1 consists of two data collections; one smaller exploratory (1a), and one larger preregistered confirmatory (1b, preregistration uploaded at https://osf.io/v2jp9/). In both, participants read vignettes describing persons engaging in helping behavior. In each vignette, we manipulated two aspects of the description: the amount of help and the type of help. Amount of help was operationalized as the size of the donation (e.g. $10 or $50 donated each month), as the number of people helped (e.g. 3 saved patients or 15 saved patients), or as the time volunteered (e.g. 2 or 10 hours per week). The large amount was usually five times larger than the small amount, but this varied some across vignettes. Type of help manipulations were different in different vignettes. We created four (2×2) conditions of each vignette by factorially manipulating both the amount of help provided (small/large amount) and the type of help (Type A = assumed to be evaluated less positively, Type B = assumed to be evaluated more positively). See Table 1

**Table 1. Summary of the vignettes included in Study 1b.** The vignettes in Study 1a were similar but not identical and e.g. showed donation amount in Swedish currency. See S1 and S2 Files for all the vignettes in all versions in all studies.

| | Type A helping (assumed to be perceived less positively) | | Type B helping (assumed to be perceived more positively) | |
|---|---|---|---|---|
| | Small amount | Large amount | Small amount | Large amount |
| **Vignette** | | | | |
| Emotional reactions (1a, 1b) | Does not get emotionally touched when observing need. Starts donating $10/month | Does not get emotionally touched when observing need. Starts donating $50/month | Gets emotionally touched when observing need. Starts donating $10/month | Gets emotionally touched when observing need. Starts donating $50/month |
| Empathy (1b) | Wants to avoid personal distress. Therefore, starts donating $6/month | Wants to avoid personal distress. Therefore, starts donating $30/month | Feels empathy for the needy. Therefore, starts donating $6/month | Feels empathy for the needy. Therefore, starts donating $30/month |
| Non-tainted altruism (1b) | Volunteers at a soup kitchen because a girl he likes works there. Volunteers 2 hours/week | Volunteers at a soup kitchen because a girl he likes works there. Volunteers 10 hours/week | Volunteers at a soup kitchen because he wants to contribute to his community. Volunteers 2 hours/week | Volunteers at a soup kitchen because he wants to contribute to his community. Volunteers 10 hours/week |
| Identified victims (1a, 1b) | Does not give money to beggars but donates $20/month to a homeless shelter | Does not give money to beggars but donates $100/month to a homeless shelter. | Gives money to beggars and donates $20/month to a homeless shelter | Gives money to beggars and donates $100/month to a homeless shelter. |
| Directness (1a, 1b) | Surgeon decides to donate part of his salary, saved 28 lives/year | Surgeon decides to donate part of his salary, saved 140 lives/year | Surgeon decides to volunteer at a refugee camp, saved 28 lives/year | Surgeon decides to volunteer at a refugee camp, saved 140 lives/year |
| Personal sacrifice (1a, 1b) | A physician volunteers in the poor areas of Rio. When not working she enjoys a luxurious life. Saved 5 lives/year | A physician volunteers in the poor areas of Rio. When not working she enjoys a luxurious life. Saved 40 lives/year | A physician volunteers in the poor areas of Rio. When not working she experiences hardships in the poor areas. Saved 5 lives/year | A physician volunteers in the poor areas of Rio. When not working she experiences hardships in the poor areas. Saved 40 lives/year |
| Keeping help private (1a, 1b) | Hung thank-you letter for a donation outside her office door. Donated $60. | Hung thank-you letter for a donation outside her office door. Donated $300 | Put thank-you letter for a donation in her office drawer. Donated $60. | Put thank-you letter for a donation in her office drawer. Donated $300. |
| Matching others (1b) | Observes acquaintance donate $2. Then donates $4 | Observes acquaintance donate $10. Then donates $20 | Observes acquaintance donate $4. Then also donates $4 | Observes acquaintance donate $20. Then also donates $20 |
| Equal helping (1a, 1b) | After winning money, he donates to 1 out of 12 requesting organizations. Donates $5,000 in total. | After winning money, he donates to 1 out of 12 requesting organizations. Donates $30,000 in total. | After winning money, he donates to all 12 requesting organizations. Donates $5,000 in total. | After winning money, he donates to all 12 requesting organizations. Donates $30,000 in total. |
| Changing amount (1b) | After donating $25/month for a year he decreases his donation to $15/month | After donating $60/month for a year he decreases his donation to $50/month | After donating $5/month for a year he increases his donation to $15/month | After donating $40/month for a year he increases his donation to $50/month |

*Note*: $1 ≈ 10SEK.

for a summary of all conditions in each vignette, and S1 and S2 Files for the complete vignettes.

## Method Study 1a

We distributed a paper and pen questionnaire at a Swedish University campus. One hundred forty-three participants (73 female, 54 male, 16 unclassified, $M_{age}$ = 24.14, $SD$ = 4.93) read seven vignettes (six reported here, one excluded vignette reported in S1 File) in one of the four conditions. We created 48 versions of the questionnaire to balance potential order or contrast effects. Participants were given a small chocolate bar.

After reading each description, participants responded to three questions about the helper; (1) What is your first impression of X? (2) Do you think that X seems unsympathetic or sympathetic? (3) Do you think that X seems like an immoral or moral person? Responses were made on a Likert scale ranging from -4 (extremely negative/unsympathetic/immoral) via 0 (neutral) to +4 (extremely positive/sympathetic/moral). The three questions correlated strongly (all $\alpha$s > .80) and were therefore aggregated into a general positive evaluation-variable.

## Method Study 1b

We initially recruited 520 American participants from Amazon Mechanical Turk (we did not ask for participants' gender or age). Participants read and responded to ten vignettes and we randomized both the order of the vignettes and which of the four conditions participants read in each vignette. Participants who completed the study were paid $2. As stated in our preregistration, we excluded participants who responded with the same number on all questions on all vignettes and participants who did not respond correctly to an embedded attention check question. After exclusions, 459 participants remained.

After each description, participants responded to the same three questions as in Study 1a with the difference that the scale went from -2 (rather bad first impression/unsympathetic/immoral) via 0 (neutral) to +5 (extremely good/sympathetic/moral). Again, we aggregated the three questions into a general positive evaluation-construct (all $\alpha$s > .80).

## Data analysis

We present results from the smaller Study 1a and the larger Study 1b together, and we focus on the effect directions and effect sizes rather than on significance levels (as suggested by e.g. [81–84], but see S3 and S5 Files for tables including $F$-statistics and $p$-values). Each vignette was analyzed separately. Mean liking for each condition in each vignette are displayed in Table 2 (and see S6 File for graphical illustrations of the results). For each vignette we conducted a 2×2 between-groups ANOVA and compared the partial eta square ($\eta_p^2$) of the two main effects (type and amount). We also conducted planned pairwise comparisons of evaluations of persons who helped with a large amount but with an assumed less sympathetic type of help (denoted A in Tables 1 and 2) against evaluations of small-amount assumed sympathetic helpers (denoted B), and report the Cohen's $d$. Parametric independent t-tests and non-parametric Mann-Whitney tests revealed almost identical results. Confidence intervals of the effect sizes (90% of the $\eta_p^2$ and 95% of the $d$) were calculated using James Uanhoro's online effect size calculators https://effect-size-calculator.herokuapp.com/. We refer to $\eta_p^2$ > .14 and $d$ > 0.8 as large; $\eta_p^2$ > .06 and $d$ > 0.5 as medium; and $\eta_p^2$ > .01 and $d$ > 0.2 as small effects [85,86]. As no strong or consistent interaction effects were found, we report them in S3 and S5 Files rather than in the main text. The raw data and code for all studies can be found in the OSF-link.

**Table 2. Means [and 95% confidence interval of the means] of liking towards helpers in each condition of each vignette in Studies 1a and 1b.**

| | Study 1a (Scale: -4 to +4) | | Study 1b (Scale: -2 to +5) | |
|---|---|---|---|---|
| *Emotional reactions vignette* | | | | |
| | *60SEK/month* | *300SEK/month* | *$10/month* | *$50/month* |
| A: Not emotionally touched | 1.48 [1.11–1.85] | 1.23 [0.89–1.57] | 2.44 [2.19–2.68] | 2.58 [2.33–2.83] |
| B: Emotionally touched | 2.09 [1.75–2.43] | 1.95 [1.61–2.30] | 3.22 [2.98–3.47] | 3.53 [3.28–3.77] |
| *Empathy vignette* | | | | |
| | | | *$6/month* | *$30/month* |
| A: Wants to avoid own distress | | | 1.79 [1.52–2.05] | 2.10 [1.84–2.35] |
| B: Feels empathy | | | 3.08 [2.82–3.33] | 3.30 [3.05–3.56] |
| *Non-tainted altruism vignette* | | | | |
| | | | *2 hours/week* | *10 hours/week* |
| A: Mixed motivation | | | 1.68 [1.44–1.91] | 2,31 [2.08–2.55] |
| B: Altruistic motivation | | | 3.36 [3.12–3.59] | 3.70 [3.46–3.94] |
| *Identified victims vignette* | | | | |
| | *100SEK/month* | *800SEK/month* | *$20/month* | *$100/month* |
| A: Gives only to shelter | 1.66 [1.35–1.96] | 2.02 [1.70–2.33] | 2.60 [2.35–2.85] | 2.93 [2.69–3.18] |
| B: Gives to identified homeless and to shelter | 2.45 [2.13–2.77] | 2.34 [2.02–2.66] | 3.31 [3.07–3.55] | 3.78 [3.54–4.02] |
| *Directness vignette* | | | | |
| | *15 saved lives* | *60 saved lives* | *28 saved lives* | *140 saved lives* |
| A: Donates part of salary (indirect help) | 2.03 [1.67–2.39] | 2.30 [1.93–2.66] | 3.52 [3.30–3.75] | 4.03 [3.81–4.26] |
| B: Volunteers at refugee camp (direct help) | 2.20 [1.84–2.56] | 2.85 [2.49–3.21] | 4.04 [3.82–4.26] | 3.91 [3.68–4.13] |
| *Personal sacrifice vignette* | | | | |
| | *11 saved lives* | *55 saved lives* | *5 saved lives* | *40 saved lives* |
| A: Enjoys life while helping | 1.97 [1.61–2.34] | 2.39 [2.03–2.75] | 3.10 [2.89–3.30] | 3.60 [3.40–3.81] |
| B: Experience hardships while helping | 2.48 [2.12–2.84] | 2.74 [2.38–3.10] | 4.17 [3.97–4.37] | 4.25 [4.04–4.45] |
| *Keeping help private vignette* | | | | |
| | *Gives 500 SEK* | *Gives 2500 SEK* | *Gives $60* | *Gives $300* |
| A: Displays donation certificate | 1.12 [0.79–1.45] | 1.50 [1.17–1.83] | 2.87 [2.62–3.12] | 2.94 [2.68–3.19] |
| B: Hides donation certificate | 1.78 [1.45–2.11] | 1.88 [1.53–2.23] | 2.68 [2.43–2.93] | 3.05 [2.79–3.31] |
| *Matching other's donation vignette* | | | | |
| | | | *Gives $4* | *Gives $20* |
| A: Surpasses acquaintance | | | 2.64 [2.37–2.91] | 2.74 [2.46–3.01] |
| B: Matches acquaintance | | | 2.68 [2.40–2.96] | 3.41 [3.14–3.68] |
| *Equal helping vignette* | | | | |
| | *72,000SEK* | *420,000SEK* | *$5,000* | *$30,000* |
| A: Gives to only one requester | 1.04 [0.66–1.41] | 1.73 [1.36–2.11] | 2.03 [1.75–2.31] | 2.46 [2.18–2.73] |
| B: Gives to all requestors | 1.70 [1.33–2.07] | 2.00 [1.62–2.38] | 2.89 [2.62–3.16] | 3.68 [3.41–3.95] |
| *Changing amount vignette* | | | | |
| | | | *to $15/month* | *to $50/month* |
| A: Decreases donation | | | 2.46 [2.21–2.71] | 2.68 [2.44–2.92] |
| B: Increases donation | | | 3.44 [3.20–3.69] | 3.56 [3.32–3.81] |

*Note*: $1 \approx$ 10SEK.

Sample size was affected by time constrains in the smaller Study 1a, but sensitivity power analyses (using G*Power, version 3.1.9.4 [87]) showed that the minimum detectable effect size for the ANOVAs ($\alpha$ = .05, power = 80%) was $\eta_p^2$ = 0.053 in Study 1a and $\eta_p^2$ = 0.017 in Study 1b. For the planned t-tests, the minimum detectable effect size ($\alpha$ = .05, power = 80%, one-tailed) was $d$ = 0.60 for Study 1a and $d$ = 0.33 for Study 1b.

**Ethics statement.** The Swedish law concerning the Ethical Review of Research Involving Humans (SFS 2003:460) serves to protect individuals and human dignity when research is conducted. In accordance with this act and based on the information on the Swedish Ethical Committee homepage, it was concluded that formal assessment was not necessary because the experimental procedure was noninvasive, did not include any deception, and because the results were analyzed on a group-level where no responses could be linked to any specific person. Furthermore, all participants were above the age of 18 and signed up willingly for participation in the specific studies. They were informed that participation was voluntary and anonymous and that they could withdraw from the experiment at any time for any reason. In order to maintain participant's anonymity and personal integrity, we did not obtain written consent.

# Results

## Emotional reactions vignette

A helper who was intensely emotionally touched when exposed to someone's plight was more positively evaluated than a helper who was not, in both Study 1a ($M_{\text{emotionally touched}} = 2.02$, $SD = 0.93$ vs. $M_{\text{not touched}} = 1.34$, $SD = 1.17$, $\eta_p^2 = .09$ [.03 − .17]), and Study 1b ($M_{\text{emotionally touched}} = 3.37$, SD = 1.31 vs. $M_{\text{not touched}} = 2.51$, $SD = 1.37$, $\eta_p^2 = .10$ [.06 − .14]).

Whether a helper donated a large or a small amount to African children did not influence evaluations much in either Study 1a ($M_{\text{300 SEK}} = 1.58$, $SD = 1.18$ vs. $M_{\text{60 SEK}} = 1.80$, $SD = 1.01$, $\eta_p^2 = .01$ [.00 − .05]), or in Study 1b ($M_{\$50} = 3.05$, $SD = 1.43$ vs. $M_{\$10} = 2.83$, $SD = 1.38$, $\eta_p^2 <$ .01 [.00 − .02]).

A helper who reacted emotionally and donated a smaller amount was more positively evaluated than a helper who donated a larger amount without any emotional reactions; $d = 0.79$ [0.31 − 1.25] in Study 1a, and $d = 0.48$ [0.22 − 0.74] in Study 1b.

## Empathy vignette (only Study 1b)

A helper who was motivated by empathy ($M_{\text{empathy}} = 3.19$, $SD = 1.31$) was much more positively evaluated than a helper motivated to avoid personal distress ($M_{\text{distress}} = 1.94$, $SD = 1.50$, $\eta_p^2 = .17$ [.11 − .21]).

Whether a helper donated a large amount ($M_{\$30/\text{month}} = 2.69$, $SD = 1.53$) or a small amount ($M_{\$6/\text{month}} = 2.44$, $SD = 1.55$) each month did not influence evaluations, $\eta_p^2 < .01$ [.00 − .03].

A helper who donated a low amount out of empathy was more positively evaluated than a helper who donated a high amount because she wanted to avoid personal distress, $d = 0.70$ [0.44–0.97].

## Non-tainted altruism vignette (only Study 1b)

A volunteer motivated by a purely altruistic motive such as helping one's community ($M_{\text{pure motive}} = 3.53$, $SD = 1.09$) was much more positively evaluated than a volunteer who was additionally motivated by spending time with a romantic crush ($M_{\text{mixed motives}} = 1.99$, $SD = 1.50$, $\eta_p^2 = .27$ [.21 − .32]).

A helper who volunteered for many hours ($M_{\text{10h/week}} = 3.01$, $SD = 1.43$) was slightly more positively evaluated than a helper who volunteered few hours ($M_{\text{2h/week}} = 2.53$, $SD = 1.56$, $\eta_p^2 = .04$ [.01 − .06]), but the effect size was considerably smaller than for type of help.

The person volunteering for few hours and being motivated by purely altruistic reasons was more positively evaluated than a helper who volunteered for many hours and was motivated by additional romantic reasons, $d = 0.77$ [0.51–1.04].

### Identified victims vignette

A helper who gave to identified homeless was more positively evaluated than a helper who gave only to a homeless shelter in Study 1a ($M_{identified\ homeless}$ = 2.40, $SD$ = 0.86 vs. $M_{only\ shelter}$ = 1.83, $SD$ = 1.03, $\eta_p^2$ = .08 [.02 − .16]). A similar effect was found in Study 1b ($M_{identified\ homeless}$ = 3.55, $SD$ = 1.24 vs. $M_{only\ shelter}$ = 2.77, $SD$ = 1.45, $\eta_p^2$ = .08 [.04 − .12]).

Whether a helper donated a large or a small amount to homeless did only marginally influence evaluations in both Study 1a ($M_{800\ SEK}$ = 2.18, $SD$ = 0.88 vs. $M_{100\ SEK}$ = 2.03, $SD$ = 1.08, $\eta_p^2$ < .01 [.00 − .04]), and Study 1b ($M_{\$100}$ = 3.37, $SD$ = 1.40 vs. $M_{\$20}$ = 2.97, $SD$ = 1.38, $\eta_p^2$ = .02 [.01 − .05]).

A helper who donated a small amount to the shelter and also gave to identified homeless was slightly more positively evaluated than a helper who donated a large amount only to the shelter $d$ = 0.46 [-0.02 − 0.94] in Study 1a, and $d$ = 0.29 [0.03–0.55] in Study 1b.

### Directness vignette

Unexpectedly, a surgeon who helped directly by volunteering in a refugee camp was only marginally more positively evaluated than a surgeon who helped indirectly by donating part of one's salary in Study 1a ($M_{direct\ volunteering}$ = 2.52, $SD$ = 1.09 vs. $M_{indirect\ donation}$ = 2.16, $SD$ = 1.11, $\eta_p^2$ = .03 [.00 − .09]), and even less so in Study 1b ($M_{direct\ volunteering}$ = 3.97, $SD$ = 1.26 vs. $M_{indirect\ donation}$ = 3.77, $SD$ = 1.20, $\eta_p^2$ < .01 [.00 - .02]).

Whether the surgeon saved many or few refugee lives only marginally influenced impressions in both Study 1a ($M_{60\ lives}$ = 2.57, $SD$ = 1.03 vs. $M_{15\ lives}$ = 2.11, $SD$ = 1.15, $\eta_p^2$ = .04 [.00 − .11]), and in Study 1b ($M_{140\ lives}$ = 3.97, $SD$ = 1.23 vs. $M_{28\ lives}$ = 3.78, $SD$ = 1.23, $\eta_p^2$ < .01 [.00 − .02]).

A surgeon who saved few patients by direct volunteering was evaluated equally good as a surgeon who saved many patients by indirect monetary donations, $d$ = -0.09 [-0.56–0.37] in Study 1a, and $d$ < .01 [-0.25–0.26] in Study 1b.

### Personal sacrifice vignette

A volunteer doctor who experienced hardships in the slums while helping was more positively evaluated than one who enjoyed a luxurious lifestyle while helping in both Study 1a ($M_{hardships}$ = 2.61, $SD$ = 1.07 vs. $M_{enjoys\ life}$ = 2.18, $SD$ = 1.13, $\eta_p^2$ = .04 [.00 −.10]), and Study 1b ($M_{hardships}$ = 4.21, $SD$ = 0.91 vs. $M_{enjoys\ life}$ = 3.35, $SD$ = 1.33, $\eta_p^2$ = .13 [.08 −.17]).

A volunteer doctor saving many lives was only slightly more positively evaluated than one saving few lives in both Study 1a ($M_{55\ lives}$ = 2.56, $SD$ = 1.02 vs. $M_{11lives}$ = 2.23, $SD$ = 1.18, $\eta_p^2$ = .02 [.00 −.08]), and in Study 1b ($M_{40\ lives}$ = 3.92, $SD$ = 1.11 vs. $M_{5\ lives}$ = 3.62, $SD$ = 1.30, $\eta_p^2$ = .02 [.00 −.04]).

A doctor who saved few lives, but experienced hardships was perceived more positively than a doctor who saved many lives but enjoyed herself in Study 1b, $d$ = 0.52 [0.26–0.79]. The same effect was weaker in Study 1a, $d$ = 0.10 [-0.37–0.56].

### Keeping help private vignette

A helper who kept her donation private (by hiding her donation certificate) was more positively evaluated than a helper who made it public in Study 1a ($M_{hiding\ certificate}$ = 1.83, $SD$ = 0.99 vs. $M_{displaying\ certificate}$ = 1.31, $SD$ = 1.03, $\eta_p^2$ = .06 [.01 −.14]). The same effect was however not found in Study 1b ($M_{hiding\ certificate}$ = 2.86, $SD$ = 1.36 vs. $M_{displaying\ certificate}$ = 2.90, $SD$ = 1.41, $\eta_p^2$ < .01 [.00 −.01]).

Whether a helper donated a large or a small amount to charity did not influence evaluations much in either Study 1a ($M_{2500 \text{ SEK}}$ = 1.68, $SD$ = 1.07 vs. $M_{500 \text{ SEK}}$ = 1.45, $SD$ = 1.01, $\eta_p^2$ = .01 [.00 − .06]), or in Study 1b ($M_{\$300}$ = 2.99, $SD$ = 1.34 vs. $M_{\$60}$ = 2.78, $SD$ = 1.42, $\eta_p^2$ < .01 [.00 − .02]).

A helper who donated a small amount in private was evaluated about equally positive as a helper who donated a large amount in public, $d$ = 0.27 [-0.20–0.73] in Study 1a and $d$ = -0.18 [-0.43–0.08] in Study 1b. It should however be noted that the directional differences were in opposite directions in these studies.

## Matching others vignette (only Study 1b)

A helper who matched donations (gave equally much as an acquaintance) was only marginally more positively evaluated ($M_{\text{matched}}$ = 3.06, $SD$ = 1.44) than a helper who surpassed donations by giving more than the acquaintance ($M_{\text{surpassed}}$ = 2.69, $SD$ = 1.58, $\eta_p^2$ = .01 [.00 − .03]).

A helper who donated a larger amount ($M_{\$20}$ = 3.08, $SD$ = 1.56) was only marginally more positively evaluated than one who donated a smaller amount ($M_{\$4}$ = 2.65, $SD$ = 1.46, $\eta_p^2$ = .02 [.00 − .04]).

A helper donating a low amount and matching someone else's donation was evaluated equally positive as one who donated a high amount and surpassed someone else's donation, $d$ = -0.04 [-0.30–0.22].

## Equal helping vignette

A helper who split his donation evenly across all requesting organizations was slightly more positively evaluated than a helper who donated to only one of several requesting organizations in Study 1a ($M_{\text{all requestors}}$ = 1.85, $SD$ = 1.13 vs. $M_{\text{only one}}$ = 1.39, $SD$ = 1.17, $\eta_p^2$ = .04 [.00 −.11]), and clearly more positively evaluated in Study 1b ($M_{\text{all requestors}}$ = 3.28, $SD$ = 1.44 vs. $M_{\text{only one}}$ = 2.24, $SD$ = 1.60, $\eta_p^2$ = .11 [.07 −.16]).

A helper who donated a large sum of money was slightly more positively evaluated than one who donated a small sum of money both in Study 1a ($M_{420K \text{ SEK}}$ = 1.86, $SD$ = 1.14 vs. $M_{72K \text{ SEK}}$ = 1.38, $SD$ = 1.15, $\eta_p^2$ = .05 [.01 −.11]) and in Study 1b ($M_{\$30K}$ = 3.08, $SD$ = 1.46 vs. $M_{\$5K}$ = 2.48, $SD$ = 1.68, $\eta_p^2$ = .04 [.02 −.07]).

A helper who donated a relatively small sum and split it across all requesting organizations was slightly more positively evaluated than one who donated a relatively high sum to a single organization in Study 1b, $d$ = 0.28 [0.02–0.54]. The same effect was however not found in Study 1a, $d$ = -0.03 [-0.49–0.44].

## Changing amount vignette (only Study 1b)

A helper who increased his initial helping amount ($M_{\text{increased}}$ = 3.50, $SD$ = 1.20) was more positively evaluated than a helper who decreased his initial donation amount ($M_{\text{decreased}}$ = 2.57, $SD$ = 1.45, $\eta_p^2$ = .11 [.07 −.16]).

A helper who donated a large amount ($M_{\$50/\text{month}}$ = 3.04, $SD$ = 1.41) was evaluated equally positive as a helper who donated a small amount ($M$ = 2.96$_{\$15/\text{month}}$, $SD$ = 1.40, $\eta_p^2$ < .01 [.00 −.02]).

The helper who changed his monthly donation from \$5 to \$15 was more positively evaluated than one who changed his monthly donation from \$60 to \$50, $d$ = 0.55 [0.29–0.82].

## Summary of Study 1

Six of the ten vignettes provided results generally supporting the person-centered approach to moral judgment, and indicating scope-insensitivity when forming impressions. Helpers who

(a) reacted emotionally (vs non-emotionally) when observing suffering; (b) were motivated by empathy (vs distress); (c) were motivated exclusively by pure altruistic (vs. mixed) motives; (d) aided (vs not aided) individual beneficiaries; (e) experienced personal hardships (vs enjoyment) while helping; and (f) increased (vs decreased) their initial level of helping, were perceived more positively even when their amount of helping was 75–80% smaller.

Four vignettes produced more diverging results. We were surprised to see that participants did not evaluate a wealthy medical doctor more positively when he volunteered (direct helping) than when he donated money (indirect helping) to aid refugees, and to see that the matching vs. surpassing manipulation had little effect. Also, Studies 1a and 1b produced rather different patterns in the keeping help private vignette and in the equal helping vignette.

## Study 2

The aims of Study 2 were threefold. First, to add additional data to the four vignettes that produced unexpected results in Study 1. Second, to explore if the observed experimental effects of type and amount of help on impressions are similar when the helpers are corporations as when they are individuals [88,89]. Previous research suggest that acts done by individuals and identical acts done by groups are perceived differently [80], but much of the large-scale helping in the world today is initiated by corporations, and one could argue that helping by corporations is more motivated by anticipated public recognition and approval than helping by individuals (strategic corporate philanthropy; [90]). Third, Study 2 tried to debias scope-insensitivity so that people would form their impressions of helpers more on the basis of the amount of help. The exploratory intervention that we tested was a simple reminder-question where the utility for the beneficiaries was made more salient. Our tentative prediction was that the utility-reminder would make participants base their impressions of helpers more on the amount of help.

## Method

A paper and pen questionnaire was distributed at a University campus to 398 Swedish participants (203 female, 188 male, 7 unclassified, $M_{age}$ = 22.00, $SD$ = 2.68). Participants read and responded to eight vignettes and as in Study 1 they read each vignette in one of the four conditions (type of helping × amount of helping). The questionnaire was constructed in 16 different versions to balance both the order of the vignettes and the vignette-condition combinations.

As in Study 1, participants responded to three questions after each vignette, and all questions used a scale ranging from -2 to +5 (same as Study 1b). However, in Study 2 we randomly allocated participants to one of two between-group conditions. Those in the control condition read the same three questions as in Study 1, but for those in the utility-reminder condition we changed the first question from "What is your first impression of X" into "How much good for the people in need does X by doing this?". The two following questions were the same in both conditions (i.e. do you think that X seems sympathetic? and moral?), and these two questions were aggregated into a general positive evaluation variable (all $r$s > .66).

Four of the eight vignettes described *individuals* who engaged in helping behavior, whereas the other four described *corporations* who did so. We chose four vignettes from Study 1 that could be re-written to describe both individual and corporate helping (albeit in different contexts), and had produced unexpected results in Study 1: (1) The directness vignette, (2) the keeping help private vignette, (3) the matching other's donation vignette and (4) the equal helping vignette. For each of these, we created one vignette describing helping by an individual and one vignette describing helping by a corporation (see S4 File for all vignettes).

**Table 3. Means [and 95% confidence interval of the means] of liking towards helpers in each condition of each vignette in Study 2 (Scale: -2 to +5).**

| | Individual helping | | Corporate helping | |
|---|---|---|---|---|
| **Directness vignette** | | | | |
| *Control* | *28 saved lives* | *140 saved lives* | *280 saved lives* | *1400 saved lives* |
| A:Helps indirectly | 2.88 [2.55–3.21] | 3.49 [3.16–3.82] | 2.65 [2.29–3.01] | 2.82 [2.46–3.18] |
| B:Helps directly | 3.36 [3.03–3.69] | 3.45 [3.12–3.78] | 3.46 [3.09–3.83] | 3.52 [3.16–3.88] |
| *Utility-reminder* | | | | |
| A:Helps indirectly | 3.23 [2.90–3.56] | 3.56 [3.23–3.89] | 2.88 [2.51–3.24] | 3.41 [3.05–3.77] |
| B:Helps directly | 3.91 [3.58–4.24] | 3.81 [3.47–4.14] | 3.84 [3.48–4.20] | 3.66 [3.30–4.02] |
| **Keeping help private vignette** | | | | |
| *Control* | *500 SEK* | *2500 SEK* | *10,000 SEK* | *50,000 SEK* |
| A: Makes helping public | 1.65 [1.28–2.03] | 1.91 [1.54–2.29] | 0.24 [−0.14–0.62] | 1.18 [0.80–1.57] |
| B: Keeps helping private | 2.53 [2.15–2.91] | 1.98 [1.61–2.36] | 2.37 [1.99–2.75] | 3.13 [2.75–3.51] |
| *Utility-reminder* | | | | |
| A: Makes helping public | 1.67 [1.30–2.05] | 2.55 [2.18–2.93] | 0.72 [0.33–1.10] | 1.12 [0.74–1.51] |
| B: Keeps helping private | 2.33 [1.95–2.71] | 2.30 [1.93–2.68] | 2.89 [2.50–3.28] | 3.56 [3.18–3.94] |
| **Matching other's donation vignette** | | | | |
| *Control* | 40 SEK | 200 SEK | *4,000 SEK* | *20,000 SEK* |
| A: Surpassing other | 0.90 [0.55–1.25] | 0.53 [0.18–0.88] | 0.31 [−0.08–0.70] | 0.67 [0.28–1.06] |
| B: Matching other | 1.12 [0.77–1.47] | 1.41 [1.06–1.76] | 0.92 [0.53–1.31] | 0.99 [0.59–1.39] |
| *Utility-reminder* | | | | |
| A: Surpassing other | 1.04 [0.69–1.39] | 0.72 [0.37–1.07] | 1.43 [1.04–1.82] | 1.25 [0.86–1.64] |
| B: Matching other | 1.21 [0.86–1.56] | 1.89 [1.53–2.24] | 0.99 [0.60–1.38] | 1.38 [0.98–1.78] |
| **Equal helping vignette** | | | | |
| *Control* | 50,000 SEK | 300,000 SEK | *400,000 SEK* | *2,000,000 SEK* |
| A: Gives to one requester | 1.50 [1.13–1.87] | 2.35 [1.98–2.72] | 1.35 [0.96–1.73] | 1.47 [1.09–1.85] |
| B: Gives to all requesters | 2.21 [1.84–2.59] | 2.48 [2.10–2.86] | 2.53 [2.15–2.91] | 2.62 [2.24–3.00] |
| *Utility-reminder* | | | | |
| A: Gives to one requester | 1.74 [1.37–2.11] | 2.46 [2.08–2.84] | 2.12 [1.74–2.50] | 2.17 [1.79–2.55] |
| B: Gives to all requesters | 2.62 [2.25–2.99] | 2.78 [2.41–3.15] | 2.70 [2.32–3.09] | 3.37 [2.99–3.75] |

## Results

In each vignette, we conducted a 2(type of help)×2(amount of help)×2(control/utility-reminder) between-group ANOVA (see Table 3 for cell means). We were primarily interested in the effect sizes of the main effects, and on the utility-reminder × amount of help interaction as this would indicate that the utility-reminder made people base their impressions more on the amount of help. As in Study 1, we also used planned independent t-tests to investigate if small-amount "sympathetic" helpers were more positively evaluated than large-amount "unsympathetic" helper (non-parametric tests provided almost identical results). Sensitivity power analyses [87] showed that the minimum detectable effect size (α = .05, power = 80%) was $\eta_p^2$ = 0.019 for the ANOVAs, and $d$ = 0.35 for the planned t-tests (one-tailed).

**Directness vignette: Individual.** There was a small main effect for type of help, $\eta_p^2$ = .02 [.01 − .06], meaning that people evaluated a surgeon volunteering at a refugee camp ($M$ = 3.63, $SD$ = 1.24) slightly more positively than a surgeon donating part of his salary to improve health care in refugee camps ($M$ = 3.29, $SD$ = 1.17).

The main effect for amount of help was $\eta_p^2$ = .01 [.00 − .04], indicating that a surgeon was similarly evaluated if he saved 140 lives ($M$ = 3.58, $SD$ = 1.23) as if he saved 28 lives ($M$ = 3.35, $SD$ = 1.19). The main effect of utility-reminder was $\eta_p^2$ = .02 [.00 − .05], meaning that the

utility-reminder slightly increased overall evaluation of helpers. There was no reminder × amount of help interaction, $\eta_p^2 < .01$.

The surgeon was evaluated equally positive if he saved 28 lives by volunteering ($M = 3.64$, $SD = 1.15$), as if he saved 140 lives by donating money ($M = 3.53$, $SD = 1.13$; $t[197] = 0.68$, $p = .498$, $d = 0.10$ [-0.18–0.37]).

**Directness vignette: Corporation.** There was a medium main effect for type of help when evaluating a corporation, $\eta_p^2 = .06$ [.03 − .10], meaning that a company who changed their main focus to produce affordable vaccines to poor people was more positively evaluated ($M = 3.62$, $SD = 1.26$) than a company who donated half of its annual profit from potency medication to buy vaccines to the poor ($M = 2.94$, $SD = 1.37$).

There was no main effect for amount of help, $\eta_p^2 < .01$ [.00 − .02], indicating that a company was evaluated equally good if it saved 1400 lives ($M = 3.35$, $SD = 1.33$) as if it saved 280 lives ($M = 3.21$, $SD = 1.38$). The utility-reminder slightly increased overall liking of helpers, $\eta_p^2 = .02$ [.00 − .04], but there was no reminder × amount of help interaction, $\eta_p^2 < .01$.

A company was evaluated more positively if it saved 280 lives directly by producing affordable vaccines ($M = 3.65$, $SD = 1.31$), than if it saved 1400 lives indirectly by donating parts of its profit ($M = 3.12$, $SD = 1.41$; $t[197] = 2.78$, $p = .006$, $d = 0.39$ [0.11–0.67]).

**Keeping help private vignette: Individual.** There was a small type of help main effect, $\eta_p^2 = .02$ [.00 − .04], meaning that a helper who put away her donation certificate ($M = 2.28$, $SD = 1.25$) was slightly more positively evaluated than a helper who hung it on the wall outside her office ($M = 1.95$, $SD = 1.47$).

There was no main effect for amount of help, $\eta_p^2 < .01$ [.00 − .02], meaning that a helper was equally positively evaluated if she donated 2500 SEK ($M = 2.19$, $SD = 1.36$) or 500 SEK ($M = 2.04$, $SD = 1.40$). The utility-reminder hardly affected evaluations of helpers, $\eta_p^2 = .01$ [.00 − .03], and there was only a very marginal reminder × amount of help interaction, $\eta_p^2 = .01$.

A person who donated 500SEK and kept the donation private ($M = 2.43$, $SD = 1.20$) was evaluated equally positive as a person who donated 2500SEK and made it public ($M = 2.23$, $SD = 1.42$, $t[196] = 1.06$, $p = .290$, $d = 0.15$ [-0.13–0.43]).

**Keeping help private vignette: Corporation.** There was a large main effect for type of help, $\eta_p^2 = .39$ [.32 − .44], meaning that people evaluate a company donating money privately ($M = 2.99$, $SD = 1.43$) much more positively than a company making their donation public ($M = 0.81$, $SD = 1.42$). There was also a small/medium main effect for amount of help in this vignette $\eta_p^2 = .06$ [.02 − .10]. People evaluated a company more positively if it donated 50,000 SEK ($M = 2.26$, $SD = 1.72$) than if it donated 10,000 SEK ($M = 1.55$, $SD = 1.80$). The utility-reminder slightly increased overall liking of helpers, $\eta_p^2 = .02$ [.00 − .04], but there was no reminder × amount of help interaction, $\eta_p^2 < .01$.

A company who donated 10,000SEK and kept the donation private ($M = 2.63$, $SD = 1.55$), was evaluated more positively than a company who donated 50,000SEK and made it public ($M = 1.15$, $SD = 1.43$, $t[195] = 6.94$, $p < .001$, $d = 0.99$ [0.70–1.29]).

**Matching other's donation vignette: Individual.** There was a small/medium main effect for type of help, $\eta_p^2 = .06$ [.02 − .10] meaning that people evaluated a helper who matched another person's donation ($M = 1.40$, $SD = 1.23$) more positively than a helper who surpassed another person's donation ($M = 0.80$, $SD = 1.32$).

There was no main effect for amount of help, $\eta_p^2 < .01$ [.00 − .01], meaning that a person donating 40SEK ($M = 1.07$, $SD = 1.28$) was evaluated equally positive as a person donating 200SEK ($M = 1.13$, $SD = 1.34$). The utility-reminder hardly affected evaluations of helpers, $\eta_p^2 = .01$ [.00 − .03], and there was no reminder × amount of help interaction, $\eta_p^2 < .01$.

A person who matched an acquaintance's donation and gave 40SEK ($M = 1.17$, $SD = 1.19$) was evaluated more positively than a person who surpassed an acquaintance's donation and gave 200SEK ($M = 0.63$, $SD = 1.26$; $t[198] = 3.12$, $p = .002$, $d = 0.44$ [0.16–0.72]).

**Matching other's donation vignette: Corporation.** There was no main effect for type of help $\eta_p^2 < .01[.00 - .02]$, suggesting that a company was evaluated equally positive if it matched another company's donation amount ($M = 1.07$, SD $= 1.39$) as if it surpassed it ($M = 0.92$, SD $= 1.50$).

There was no main effect for amount of help, $\eta_p^2 < .01[.00 - .02]$, meaning that a company was evaluated equally positive if it donated 20,000 SEK ($M = 1.07$, $SD = 1.45$) as if it donated 4,000 SEK ($M = 0.91$, $SD = 1.44$). The utility-reminder increased overall evaluations of helpers $\eta_p^2 = .04$ [.01 − .07], but there was no reminder × amount of help interaction, $\eta_p^2 < .01$.

A company matching a rival company's donation and giving 4,000 SEK ($M = 0.96$, $SD = 1.36$) was rated equally positively as a company surpassing a rival company and giving 20,000 SEK ($M = 0.96$, $SD = 1.47$; $t[198] = -0.03$, $p = .980$, $d = -0.00$ [-0.28–0.27]).

**Equal helping vignette: Individual.** There was a small main effect for type of help $\eta_p^2 = .04$ [.01 − .07], meaning that people evaluate a donor who splits his donation across all requesting organizations ($M = 2.53$, $SD = 1.32$) slightly more positively than a donor who donates to only one of the requesting organization ($M = 2.01$, $SD = 1.42$).

There was also a small main effect for amount of help $\eta_p^2 = .03$ [.01 − .07], meaning that people evaluate a 300,000 SEK donor ($M = 2.52$, $SD = 1.38$) slightly more positively than a 50,000SEK donor ($M = 2.02$, $SD = 1.37$). The utility-reminder hardly affected liking of helpers, $\eta_p^2 = .01$ [.00 - .03], and there was no reminder × amount of help interaction, $\eta_p^2 < .01$.

A person splitting a 50,000 SEK donation across all requesters was evaluated equally positive ($M = 2.42$, $SD = 1.28$) as a person giving 300,000 SEK to a single requesting organization ($M = 2.40$, $SD = 1.39$; $t[196] = 0.08$, $p = .936$, $d = 0.01$ [-0.26–0.29]).

**Equal helping vignette: Corporation.** There was a medium main effect for type of helping, $\eta_p^2 = .13$ [.08 − .18], meaning that people evaluated a bank who splits its charitable donation across all requesting causes ($M = 2.81$, $SD = 1.43$) more positively than a bank who supports only one of the requesting causes ($M = 1.78$, $SD = 1.36$).

There was hardly any effect for amount of help, $\eta_p^2 = .01$ [.00 − .03], meaning that people evaluated a bank equally positive if it donated 0.4 million SEK as if it donated 2 million SEK to charitable causes. There was again a main effect for reminder, $\eta_p^2 = .05$ [.02 − .09], meaning that the utility-question improved general liking of the bank, but there was no reminder × amount of help interaction, $\eta_p^2 < .01$.

A bank splitting its 0.4 million SEK donation across all requesting causes ($M = 2.62$, $SD = 1.49$) was evaluated more positively than a bank donating 2 million SEK to a single cause ($M = 1.82$, $SD = 1.41$; $t[197] = 3.87$, $p < .001$, $d = 0.55$ [0.27–0.83]).

## General discussion

Over three studies and ten vignette-types, we tested how impressions of helpers change as the amount of help and ten different types of help (motivational or situational aspects) are manipulated. The take-home message is that the type of help predicts impressions of helpers much better than the amount of help, but that some situational and motivational aspects of helping, influence impressions much more than others.

### Six vignettes with predicted results

In six of the ten vignettes in Study 1 (the emotional reactions, empathy, non-tainted altruism, identified beneficiaries, changing amounts and personal sacrifice vignette), we found medium

or large effect sizes for the type of help manipulations and no or small effect sizes for the amount of help manipulations. In these vignettes, people evaluated low-amount helpers more positively than high-amount helpers who did e.g. 500% more good, but were differently motivated, or helped under different circumstances.

This is in line with predictions and clearly illustrates scope-insensitivity in an impression formation context. The type of help manipulations with the largest effect sizes were (1) whether the helper had non-tainted (pure) motives for volunteering or if the helping in part was motivated by a wish to spend time with a romantic crush [54], and (2) whether the helper was motivated by empathy rather than by distress/guilt avoidance [53]. In both these vignettes, we manipulated underlying motivations of the helper directly (by stating them), whereas we, in most other vignettes, manipulated observable aspects of the situation. Situational aspects can of course be used to draw inferences about the underlying motives of helping, and in the real world we have rarely access to the actual motives of helpers but have to rely on observations and past experiences. A recent study argues that people tend to evaluate individuals who are doing ambiguous acts towards others negatively [91], and although helping is a positive behavior in its core, it can easily become ambiguous when there is a risk that the helping is driven by the "wrong" motives.

This, we argue, supports the person-centered approach to moral judgments proposed by Uhlmann et al. [13,14,25,26]. Simply put, observers do not evaluate prosocial behavior in itself but rather the assumed underlying processes and motives of prosociality.

## Four vignettes with mixed results

The results from the four remaining vignettes did not render as easily interpretable results in Study 1 so we scrutinized them further in Study 2, where we also tested impressions of corporations as well as individuals engaging in helping behavior, and made the utility of helping more salient by adding a reminder question before participants reported their impressions.

The directness manipulation for individual helpers did not improve impressions to the expected extent. A direct helper (e.g. a surgeon volunteering at a refugee camp) was not perceived as more sympathetic than an indirect helper (e.g. a surgeon donating part of his salary to pay others to work at a refugee camp). One reason for the surprisingly small effect might be that this vignette elicited the most positive impressions overall. Participants (who read multiple vignettes) might have always perceived the helper in this vignette as the relatively most sympathetic, as he was implicitly compared against the helpers described in the other vignettes. If this vignette was tested in isolation or if its content was better matched against the other vignettes, we might have found the effect of directness. Importantly, when testing the effect in a corporate context (using a quite different vignette) we did find a medium sized effect of directness.

The equal helping manipulation did influence impressions to some extent (small or medium effect sizes), but the amount of help manipulation also influenced impressions relatively much in this vignette. Still, lottery-winners who gave away a smaller amount of money to all requesting charities were at least as positively evaluated as those who gave away a 5–6 times larger sum to a single organization. Moreover, when the helper was a bank rather than an individual (in Study 2), impressions were much more influenced by equal distribution than by the actual amount given away.

The keeping private manipulation provided mixed results that might be attributed to cultural differences. In the two studies conducted in Sweden (Studies 1a and 2), but not in the study conducted in the US (Study 1b), we found that actively displaying one's donation certificate, to some extent reduced liking of the helper. Whereas the US is traditionally considered a

country where self-promotion and approach (vs. avoidance) goals are seen as virtues [92], Scandinavia is said to sometimes employ "the law of Jante" meaning that it is frowned upon to perform better than others, and especially to take pride in ones accomplishments [93]. To show one's donation-certificate for others might be seen as breaking the law of Jante among Swedes, but as a virtuous attempt to inspire others to follow ones lead in the US.

The matching vignette did not produce any main effects in Study 1b but people who matched an acquaintance's donation (gave an equal amount) were perceived more positively than a person who surpassed it (gave a higher amount) in Study 2. Cultural differences (as mentioned above) could have played a role in this vignette as well. It is also worth noting that this was the only vignette where we found the same interaction effect in both studies (See S3 and S5 Files for detailed information about the interaction effects). A larger amount donated improved impressions when matching another's donation but almost worsened impressions when surpassing another's donation. People might perceive a helper who observes an acquaintance donate a reasonably large amount ($10), and then surpass that by donating $20 as motivated by an urge to "win" rather than by an urge to help [74]. We also note that organizations who surpass each other are perceived as no worse than organizations who match each other's donations, presumably because corporations always compete, and it is better if they compete in charity than if they compete in profit.

## Did the utility-reminder debias scope-insensitivity?

The utility-reminder in Study 2 did not produce any robust reminder × amount of help interaction effect in any of the eight vignettes so we consider this attempt to debias scope-insensitivity a failure. However, in five of the eight vignettes we found that impressions of helpers were slightly more positive for participants who first responded to the utility-reminder question "How much good for the people in need does X by doing this?" than for participants who first responded to the control question "What is your first impression of X?". It is possible that the first question changed the meaning of the two forthcoming questions that we used to assess impressions of the helper (i.e. "how sympathetic" and "how moral" does X seem to you?). Importantly, the utility-reminder made people more favorable toward all helpers, not only toward the high-amount helpers.

## Limitations and future research

As previously mentioned, these studies were conducted with a wide rather than narrow focus. The upside of this approach is that it unifies and summarizes previous research about impression formation of helpers, and that it can be used as a starting point for future research on this topic. The downside is that there is more research to do before we can fully understand the boundary conditions and psychological mechanisms involved in impression formation of helpers. It would be possible (and recommendable) to devote one article to each of the ten vignettes included here and test each effect in a variety of contexts, using different types of manipulations and outcome variables (e.g. more specified person perception variables, behavioral measures such as willingness to sponsor or reward the helper, and evaluation of actions rather than individuals in order to test act-person dissociations [26]), systematically testing moderators and meditators of the effects, and investigating other ways to nudge people into being more sensitive to scope when forming (and expressing) impressions about helpers.

A methodology concern could be that the background information was different in the different vignettes. To clarify, the purpose of this study was not to compare the different vignettes directly against each other, but to compare how the amount and type of help manipulations

influenced impressions of helpers within each vignette (where background information was held constant). Still, we acknowledge that specific background information (e.g. gender or occupation of the helper) might interact with specific type of help manipulations when forming impressions.

Relatedly, we manipulated amount of help differently in the different vignettes. Seven vignettes manipulated charitable giving (small/large donation), two manipulated outcomes (few/many saved lives), and one manipulated hours of volunteer work (short/long time). Johnson [94] suggest that money and time (input amount) and lives saved (output amount) are perceived differently so future studies might benefit from keeping the "amount unit" constant across vignettes. Also, although we kept the proportional difference between the "low" and "high" amount conditions relatively equal in the different vignettes (usually X vs. 5X), one can surely argue that the *perceived* amount ratio between $0.5 and $2.5 is smaller than that between $60 and $300 even if the *actual* amount ratio is identical.

Crucially, all three studies in this paper were conducted in separate evaluation meaning that each participant only read and evaluated one condition of each vignette [95,96]. We are currently investigating whether people will base their impressions more on the amount of help (and less on the type) when they can compare e.g. an emotional low-amount helper against a non-emotional high-amount helper (i.e. joint evaluation). Some previous research suggest that joint evaluation makes people more sensitive to scope (e.g. amount) when making helping decisions [97], but at the same time people form negative impressions of those who choose to help more outgroup members rather than fewer friends and family members [98,99]. It remains to be seen which situational and motivational aspects of helping that will trump amount of help when forming impressions of helpers in joint evaluation.

Another aspect worth adding in future studies on impression formation is individual and cultural differences. It is very possible that some groups of observers are more influenced by the amount of help than others, and that one type of help-manipulation influences impressions much for some observers but not for others. To exemplify, familiarity and expertise can increase the evaluability of numerical differences [96] and donors that are highly committed to a specific cause, or very prosocial in general, might base more of their impressions of helpers on the amount of help given than on the type of help [100,101].

## Conclusion

Humans have always formed impressions about each other, but one can argue that it is even more important in the current era, when much more of our behavior, written text and spoken word is posted online and used by others to form impressions of us. Humans try to avoid behavior that can elicit a negative impression, but also try to engage in behavior that can make observers form more positive impressions of them. Helping is one such behavior, and in this article, we investigated impressions of helpers from a broad perspective and empirically tested which aspects of help behavior that predict impressions. The take-home message is that the amount (or the consequences) of help do not influence impression of helpers much, but that many situational and motivational aspects of helping does. Especially the perceived purity of the motives, and the accompanying appropriate emotional reactions seem central for helpers who aim to be positively evaluated.

This implies that it is primarily the helpers that do not care about how they are perceived who will elicit the most positive impressions when helping. Whereas the paradox of happiness suggests that a person who actively pursues happiness cannot become happy [102], we suggest a "paradox of praise", meaning that the person who helps in order to be liked will have trouble doing so (unless they are really good actors).

## Supporting information

**S1 File. All vignettes included in Study 1a.**
(DOCX)

**S2 File. All vignettes included in Study 1b.**
(DOCX)

**S3 File. Main effects and interaction effects in Study 1.** (F-statistics and significance-values).
(DOCX)

**S4 File. All vignettes included in Study 2.**
(DOCX)

**S5 File. Main effects and interaction effects in Study 2.** (F-statistics and significance-values).
(DOCX)

**S6 File. Graphical illustrations of the results from each vignette in all studies.**
(DOCX)

## Author Contributions

**Conceptualization:** Arvid Erlandsson, Mattias Wingren.

**Data curation:** Arvid Erlandsson, Mattias Wingren.

**Formal analysis:** Arvid Erlandsson, Mattias Wingren, Per A. Andersson.

**Funding acquisition:** Arvid Erlandsson.

**Investigation:** Arvid Erlandsson, Per A. Andersson.

**Methodology:** Arvid Erlandsson, Mattias Wingren.

**Project administration:** Arvid Erlandsson, Mattias Wingren.

**Supervision:** Arvid Erlandsson.

**Writing – original draft:** Arvid Erlandsson.

**Writing – review & editing:** Mattias Wingren, Per A. Andersson.

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
