## [Decision Letter · Decision Letter 0]

26 Oct 2020

PONE-D-20-26816

Type and amount of help as predictors for impression of helpers

PLOS ONE

Dear Dr. Erlandsson,

Thank you for submitting your manuscript to PLOS ONE. After careful consideration, we feel that it has merit but does not fully meet PLOS ONE’s publication criteria as it currently stands. Therefore, we invite you to submit a revised version of the manuscript that addresses the points raised during the review process.

We look forward to receiving your revised manuscript.

Kind regards,

Valerio Capraro

Academic Editor

PLOS ONE

Additional Editor Comments:

I have now received two reviews from two experts in the field. Both reviewers like the paper but suggest several major revisions. Therefore, I would like to invite you to revise your work following the reviewers' comments. I am looking forward for the revision.

Journal Requirements:

Reviewers' comments:

Reviewer's Responses to Questions

**Comments to the Author**

1. Is the manuscript technically sound, and do the data support the conclusions?

Reviewer #1: Yes

Reviewer #2: Partly

2. Has the statistical analysis been performed appropriately and rigorously? 

Reviewer #1: Yes

Reviewer #2: Yes

3. Have the authors made all data underlying the findings in their manuscript fully available?

Reviewer #1: Yes

Reviewer #2: Yes

4. Is the manuscript presented in an intelligible fashion and written in standard English?

Reviewer #1: Yes

Reviewer #2: Yes

5. Review Comments to the Author

Reviewer #1: Thank you for giving me the possibility to review the manuscript “Type and amount of help as predictors for impression of helpers”. The topic is socially relevant, the manuscript is well written and the argumentation is consistent. Giving the number of experimental conditions, I find that authors structured very well the paper, in such a way that is easy to follow. However, there are some concern that could be improved for a higher quality in the final document.

1. Research question and literature review, argumentation of hypothesis are well explained and structured, in line with the hypothesis proposed by the authors. On page 6, line 114, the authors explain that the “types of helping were chosen to represent a wide array of motivational and situational aspects…”. How were they chosen? How was the representativeness decided or what was the criteria?

2. The potential differences in experimental effects between individual and groups are only introduced on page 11, when the studies are presented. The authors should previously argument the decision of introducing it.

3. Study 1a, page 12, line 266, is not clear why 6 out of 7 vignettes were reported in the manuscript.

4. For the vignette “matching others”, and “changing amount” study 1b, the results might be affected by an anchoring effect. The authors could explain how was that considered or not?

5. If for the Study 1b, the authors calculated the sample power, given the considerable sample size, it should also be calculated for the study 1b, where the sample size was smaller.

6. For study 1a it is not indicated what was the reward of the participants in the study, as for study 1b, it is specified that the participants received 2$.

7. ANOVA analysis relies on normality assumptions. The authors should provide a test of normality to justify such tests.

8. I recommend including the direct effects and the interaction effect in a table as supplementary file. It would be much easier to flag the relevant results for the three studies.

9. The sentence starting at line 502 is not clear. Please rephrase.

10. There is no information what is the level of prosociality or the previous experience on prosocial behaviours of the participants in the study. When studying perceived situation, it is an important factor to control for.

11. Formal aspects: in some cases, the brackets are not closed ().

12. References list are not in the PlosOne format (Vancouver style). Please revise.

Reviewer #2: This paper examines how type and amount of help contribute to the impression that third parties have about helpers. This is an interesting issue. The paper could make progress in understanding how humans make evaluations about other’s morality. However, I have some concerns that I describe below.

1- In the first part of the manuscript I miss more information about the specific contribution of the study. What is the difference between this study and previous research efforts? I also miss convincing initial arguments about the relevance of selected variables.

2- As the authors reported on page 4, first paragraph, research has demonstrated that negative behaviors are more influential in explaining moral judgments than positive ones. However, they focus on positive behaviors (helping) without providing a convincing explanation or justification in terms of contribution to knowledge. Why, if the negative has more influence, does the manuscript focus on the positive?

3- Regarding the influence of “amount” on impression, there are previous studies that reported lack of significant impact (pages 5-6). Thus, what is the contribution of the current study? More specific information is needed to evaluate the current contribution.

4- I also have doubts about the contribution of the study to previous literature in other propositions: the role of emotions (pages 6-7); empathy (page 7); pure-mixed motives (pages 7-8); direct vs indirect help (pages 8-9); costs (page 9): public vs. private (pages 9-10); norms (page 10); equal vs. unequal allocations (pages 10-11); and upward vs. downward (page 11). In all these propositions, there are previous research efforts. It is not clear what is the specific contribution of the manuscript. On page 11 (first full paragraph), the authors explain their rationale. It is, however, a short paragraph. There is no significant information about the contribution to knowledge.

5- One important concern is related to the research design. The authors carried out three studies with different combination of vignettes. There is no information about the rationale underlying this design. Why? What is the intention of authors with this design? Why does this design allow advances in knowledge? Some aspects are provided (e.g., on page 26) but more effort is necessary to provide significant and substantive arguments related to the decision-making.

6- Thus, I urge the authors to describe the contribution in detail (comparing the current manuscripts with previous ones); explain the logic underlying the decisions about selected variables; and explain the decision-making about the design (why these studies and their format) in terms of contribution to knowledge and solid response to the research questions.

6. PLOS authors have the option to publish the peer review history of their article (what does this mean?). If published, this will include your full peer review and any attached files.

Reviewer #1: No

Reviewer #2: No

---

## [Author Response · Author response to Decision Letter 0]

6 Nov 2020

Thank you for the insightful reviews. We have responded to the points raised by the reviewers below and modified the manuscript accordingly. 

Reviewer #1: Thank you for giving me the possibility to review the manuscript “Type and amount of help as predictors for impression of helpers”. The topic is socially relevant, the manuscript is well written and the argumentation is consistent. Giving the number of experimental conditions, I find that authors structured very well the paper, in such a way that is easy to follow. However, there are some concern that could be improved for a higher quality in the final document.

1. Research question and literature review, argumentation of hypothesis are well explained and structured, in line with the hypothesis proposed by the authors. On page 6, line 114, the authors explain that the “types of helping were chosen to represent a wide array of motivational and situational aspects…”. How were they chosen? How was the representativeness decided or what was the criteria?

RESPONSE: The reason for including these specific “type of help-manipulations” was not grounded in a specific theory but rather inspired by existing literature in combination with personal experiences. Rather than focusing on a single manipulation, we wanted to include several diverse type of help-manipulations (including both motivational and situational). But at the same time, we do not argue that the included manipulations represent an exhaustive list. This has been clarified under the header “ten types of helping” 

It should be noted that variations of some of our type of help-manipulations (e.g. Emotional reactions, Non-tainted altruism) have been tested in previous studies, and for these manipulations our results should be seen as conceptual replications. In contrast, we are, to our knowledge, the first to empirically test how some of these manipulations influence impression formation of helpers (e.g. empathy, identified victim, matching others and changing amount). We are now clarifying which manipulations that we are the first to test (as far as we know). 

2. The potential differences in experimental effects between individual and groups are only introduced on page 11, when the studies are presented. The authors should previously argument the decision of introducing it.

RESPONSE: We now introduce the idea that helpers can be individuals but also corporations in the opening paragraph, but as the individual vs group distinction is rather peripheral in this manuscript , we wish to not dwell on it too much early in the manuscript. We can certainly move the text about differences between individuals and groups from Page 26 (intro to Study 2) to the introduction section if necessary, but we hope that simply reminding readers that helping can be done by both individuals and corporations is sufficient. 

3. Study 1a, page 12, line 266, is not clear why 6 out of 7 vignettes were reported in the manuscript.

RESPONSE: The reason for not including this vignette in the main manuscript is that it was not included in the more extensive and well-powered Study 1b (as it did not render any effects in the smaller Study 1a). We believe that including this vignette in the main text would impair readability, but we now include it in the online supplementary material S1 (both stimuli material and results).

4. For the vignette “matching others”, and “changing amount” study 1b, the results might be affected by an anchoring effect. The authors could explain how was that considered or not?

RESPONSE: This is a good point. Upon reflection, we would probably argue that these type of help-manipulations in many ways are based on anchor effects. In these (but not in the other manipulations), the type of help manipulation is dependent on the relation between two amounts, namely acquaintance’s donation vs helper’s donation in “matching others”, and earlier donation vs current donation in “changing amount”. In other words, rather than saying that the results might be affected by an anchoring effect, we argue that an anchoring effect (in its broad definition) is included in these type of help-manipulations. We now mention the anchoring effect when introducing the changing amount manipulation.

5. If for the Study 1b, the authors calculated the sample power, given the considerable sample size, it should also be calculated for the study 1b, where the sample size was smaller.

RESPONSE: We now report sensitivity power analyses for all studies. 

6. For study 1a it is not indicated what was the reward of the participants in the study, as for study 1b, it is specified that the participants received 2$.

RESPONSE: We have now clarified that participants in Study 1a received a small chocolate bar for their participation.

7. ANOVA analysis relies on normality assumptions. The authors should provide a test of normality to justify such tests.

RESPONSE: R1 is very right that factorial ANOVA relies on several assumptions such as normality and homoscedasticity of variances. Real data rarely fulfill all these assumptions to 100%, but importantly not all assumption-violations poses a problem in all situations. 

Regarding normality, the central limit theorem argue that we can assume normality regardless of the shape of the sample data when having sufficiently big samples (as we have in at least Study 1b and 2). Also, Field (page 235 and 248) argue against using e.g. Kolmogorov-Smirnov tests to test normality in large sample-sizes as minimal violations of normality will turn out significant. 

Regarding homoscedasticity, violating this assumption matters only if one has unequal group sizes (Field, page 259). If group-sizes are equal this assumption can be ignored. All group sizes in this study are equal or roughly equal. 

One approach when dealing with simple effects is to use both parametric and non-parametric tests. To make sure that our conclusions are not artefacts of the analyses, we conducted non-parametric Mann Whitney tests for all the planned comparisons in all three studies (high-amount sympathetic helper vs. low-amount non-sympathetic helper). All results found in the independent t-tests were also found in the Mann-Whitney tests, suggesting that the results are sufficiently robust. We now mention this in the manuscript. 

Raw data from all studies are freely available, so any reader can conduct more advanced tests, but we prefer to report result from analyses that most readers are familiar with. 

Field A. Discovering Statistics Using IBM SPSS Statistics 5th ed. Sage; 2018.

8. I recommend including the direct effects and the interaction effect in a table as supplementary file. It would be much easier to flag the relevant results for the three studies.

RESPONSE: Good point! We have added tables as requested in supplementary files S3 (Study 1a and 1b) and S5 (Study 2). We have also plotted the results from each vignette from each study in a figure a show these graphical illustrations in supplementary file S6. We are prepared to move the tables and/or the figures to the main text if you or the editor thinks it would make the paper easier to digest. 

9. The sentence starting at line 502 is not clear. Please rephrase.

RESPONSE: We have rephrased this sentence and hope it is clearer now. 

10. There is no information what is the level of prosociality or the previous experience on prosocial behaviours of the participants in the study. When studying perceived situation, it is an important factor to control for.

RESPONSE: This is a good point. We did not measure participants’ level of prosociality in this study, but we agree that it is possible that participants’ prosociality could interact with the type of help- and amount-manipulations so that highly prosocial participants would base their impressions more on amount whereas less prosocial participants would base their impressions more on type of help. We now mention individual differences as a future direction is the general discussion section. 

11. Formal aspects: in some cases, the brackets are not closed ().

RESPONSE: Thank you, we have fixed this in the new version. Please let us know if we have missed it somewhere. 

12. References list are not in the PlosOne format (Vancouver style). Please revise.

RESPONSE: Thank you, we have fixed this in the new version. 

 

Reviewer #2: This paper examines how type and amount of help contribute to the impression that third parties have about helpers. This is an interesting issue. The paper could make progress in understanding how humans make evaluations about other’s morality. However, I have some concerns that I describe below.

RESPONSE: In most comments raised by Reviewer 2, the main concern seems to be what the “specific contribution” of the current study is. We explain what we think about this paper’s contribution below. As we realize than others than Reviewer 2 might have similar concerns, we now spell out our opinion about the contribution of this manuscript on page 12. 

To be clear and perfectly honest, the main quality of this paper is not its novelty or its unique empirical contribution, but rather that it represents rigorous basic experimental social psychology research that is theoretically grounded in the person-centered approach to moral judgment, and can be applied to societal issues (e.g. charitable giving, volunteering and “effective altruism”). Over and above contributing to the collected moral impression formation research (which we describe to the best of our ability), we think that this paper summarizes existing research related to this topic in a systematic way, and place it under a unifying umbrella term. Therefore, we believe that this paper can be a natural starting point and inspiration for future research about impression formation of helpers. 

Importantly, Plos One distinguish itself by being a journal where rigorously conducted basic research can be published even when novelty is limited. This is one contributing reasons for submitting this manuscript to Plos One. Reviewer 2 is probably right to question the “specific contribution” of the conducted studies, but we argue that empirical studies that test phenomena found in previous studies, contribute to the collected knowledge and are valuable, even if the contribution is not “specific”. 

1- In the first part of the manuscript I miss more information about the specific contribution of the study. What is the difference between this study and previous research efforts? I also miss convincing initial arguments about the relevance of selected variables.

RESPONSE: The reason for including these specific “type of help-manipulations” was not grounded in a specific theory but rather inspired by existing literature in combination with personal experiences. Rather than focusing on a single manipulation, we wanted to include several diverse type of help-manipulations (including both motivational and situational). But at the same time, we do not argue that the included manipulations represent an exhaustive list. This has been clarified under the header “ten types of helping” 

2- As the authors reported on page 4, first paragraph, research has demonstrated that negative behaviors are more influential in explaining moral judgments than positive ones. However, they focus on positive behaviors (helping) without providing a convincing explanation or justification in terms of contribution to knowledge. Why, if the negative has more influence, does the manuscript focus on the positive?

RESPONSE: Our main reason for focusing on the positive (impression of helpers) rather than the negative (impression of harmers) is first and foremost that judgments about positive moral behavior and positive moral actors have been less researched than negative moral behavior and negative moral actors. In fact, previous review articles have asked for more research related to moral praise (in contrast to moral blame). See e.g. Bartels DM, Bauman CW, Cushman FA, Pizarro DA, McGraw AP. Moral Judgment and Decision Making. The Wiley Blackwell Handbook of Judgment and Decision Making. 2014:478-515.

Saying that negative behaviors influence impression formation more than positive behaviors does not imply that positive behaviors does not influence impressions at all. We now clarify this more explicitly in the opening pages. 

3- Regarding the influence of “amount” on impression, there are previous studies that reported lack of significant impact (pages 5-6). Thus, what is the contribution of the current study? More specific information is needed to evaluate the current contribution.

RESPONSE: We report a few studies that investigated how impressions (or related constructs) are affected when the amount of help is higher or lower. However, all but one of these studies investigated prosocial behavior in economic games rather than in charitable giving. We think that mentioning these articles is a more honest approach than pretending they did not exist (which could be done if we wanted to portray our research as more novel than it is). We are not the first to investigate if “amount of help” influence impressions, but our studies can add to the existing knowledge. Again, even if the “specific contribution” is limited, we argue that this research is still a contribution that is worth publishing. 

4- I also have doubts about the contribution of the study to previous literature in other propositions: the role of emotions (pages 6-7); empathy (page 7); pure-mixed motives (pages 7-8); direct vs indirect help (pages 8-9); costs (page 9): public vs. private (pages 9-10); norms (page 10); equal vs. unequal allocations (pages 10-11); and upward vs. downward (page 11). In all these propositions, there are previous research efforts. It is not clear what is the specific contribution of the manuscript. On page 11 (first full paragraph), the authors explain their rationale. It is, however, a short paragraph. There is no significant information about the contribution to knowledge.

RESPONSE: Our initial response about the importance of “specific contribution” applies here as well. 

It should be noted that variants of some of our help-manipulations have been tested in previous studies (e.g. Emotional reactions, Directness and Non-tainted altruism), and for these manipulations our results should be seen as conceptual replications that adds to the existing knowledge. 

In contrast, we are, to our knowledge, the first to empirically test how some of these manipulations influence impression formation of helpers (e.g. empathy, identified victim, matching others, and changing amount). We have now clarified which manipulations that we are the first to test in an impression formation context. 

5- One important concern is related to the research design. The authors carried out three studies with different combination of vignettes. There is no information about the rationale underlying this design. Why? What is the intention of authors with this design? Why does this design allow advances in knowledge? Some aspects are provided (e.g., on page 26) but more effort is necessary to provide significant and substantive arguments related to the decision-making.

RESPONSE: See above

6- Thus, I urge the authors to describe the contribution in detail (comparing the current manuscripts with previous ones); explain the logic underlying the decisions about selected variables; and explain the decision-making about the design (why these studies and their format) in terms of contribution to knowledge and solid response to the research questions.

RESPONSE: See above

---

## [Decision Letter · Decision Letter 1]

27 Nov 2020

Type and amount of help as predictors for impression of helpers

PONE-D-20-26816R1

Dear Dr. Erlandsson,

We’re pleased to inform you that your manuscript has been judged scientifically suitable for publication and will be formally accepted for publication once it meets all outstanding technical requirements.

Kind regards,

Valerio Capraro

Academic Editor

PLOS ONE

Additional Editor Comments (optional):

Reviewers' comments:

Reviewer's Responses to Questions

**Comments to the Author**

1. If the authors have adequately addressed your comments raised in a previous round of review and you feel that this manuscript is now acceptable for publication, you may indicate that here to bypass the “Comments to the Author” section, enter your conflict of interest statement in the “Confidential to Editor” section, and submit your "Accept" recommendation.

Reviewer #1: All comments have been addressed

Reviewer #2: All comments have been addressed

2. Is the manuscript technically sound, and do the data support the conclusions?

Reviewer #1: Yes

Reviewer #2: Yes

3. Has the statistical analysis been performed appropriately and rigorously? 

Reviewer #1: Yes

Reviewer #2: Yes

4. Have the authors made all data underlying the findings in their manuscript fully available?

Reviewer #1: Yes

Reviewer #2: Yes

5. Is the manuscript presented in an intelligible fashion and written in standard English?

Reviewer #1: Yes

Reviewer #2: Yes

6. Review Comments to the Author

Reviewer #1: I have now revised the manuscript with the suggested changes inserted. I would like to thank the authors for responding and including all the questions addressed during the revision process.

I believe the manuscript now is more clear to the readership and some important issues have been address in the methodology part.

On my behalf, the manuscript is ready to be published in its actual form.

Reviewer #2: Dear authors, thank you for considering my comments and for the opportunity to read this manuscript. I appreciate the contribution but also the solid efforts to replicate previous findings.

7. PLOS authors have the option to publish the peer review history of their article (what does this mean?). If published, this will include your full peer review and any attached files.

Reviewer #1: No

Reviewer #2: No

---

## [Editor Report · Acceptance letter]

4 Dec 2020

PONE-D-20-26816R1 

Type and amount of help as predictors for impression of helpers 

Dear Dr. Erlandsson:

I'm pleased to inform you that your manuscript has been deemed suitable for publication in PLOS ONE. Congratulations! Your manuscript is now with our production department. 

Kind regards, 

on behalf of

Dr. Valerio Capraro 

Academic Editor

PLOS ONE